# Fairness Evaluation with Item Response Theory

## Abstract

Item Response Theory (IRT) has been widely used in educational psychometrics to assess student ability, as well as the difficulty and discrimination of test questions. In this context, discrimination specifically refers to how effectively a question distinguishes between students of different ability levels, and it does not carry any connotation related to fairness. In recent years, IRT has been successfully used to evaluate the predictive performance of Machine Learning (ML) models, but this paper marks its first application in fairness evaluation. In this paper, we propose a novel Fair-IRT framework [1] to evaluate a set of predictive models on a set of individuals, while simultaneously eliciting specific parameters, namely, the ability to make fair predictions (a feature of predictive models), as well as the discrimination and difficulty of individuals that affect the prediction results. Furthermore, we conduct a series of experiments to comprehensively understand the implications of these parameters for fairness evaluation. Detailed explanations for item characteristic curves (ICCs) are provided for particular individuals. We propose the flatness of ICCs to disentangle the unfairness between individuals and predictive models. The experiments demonstrate the effectiveness of this framework as a fairness evaluation tool. Two real-world case studies illustrate its potential application in evaluating fairness in both classification and regression tasks. Our paper aligns well with the Responsible Web track by proposing a Fair-IRT framework to evaluate fairness in ML models, which directly contributes to the development of a more inclusive, equitable, and trustworthy AI.

## CCS Concepts

• **Information systems** → **Decision support systems**; • **Computing methodologies** → *Machine learning*; • **Applied computing** → Law, social and behavioral sciences.

## Keywords

Fairness Evaluation, Item Response Theory

**ACM Reference Format:**
Anonymous Author(s). 2018. Fairness Evaluation with Item Response Theory. In *Proceedings of Make sure to enter the correct conference title from your rights confirmation emai (Conference acronym ’XX)*. ACM, New York, NY, USA, 12 pages. https://doi.org/XXXXXXX.XXXXXXX

[1]The source code can be found at https://anonymous.4open.science/r/Fair-IRT-B17C.

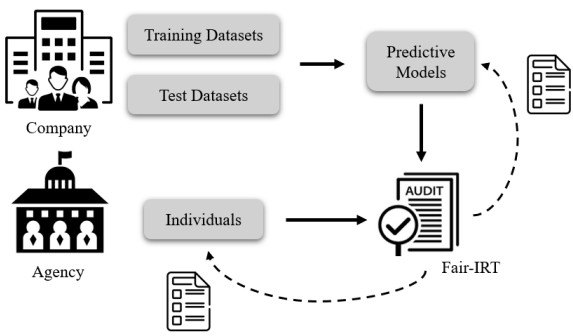

**Figure 1: The general scenario in fairness evaluation. The dashed line denotes the two analysis directions: one for individuals and another for predictive models.**

## 1 Introduction

Item Response Theory (IRT) is a framework that originated in the mid-20th century and is primarily applied in psychometrics. It aims to characterise both items and respondents through the analysis of responses [11, 14, 17]. In recent years, IRT has been proposed to evaluate predictive performance in machine learning (ML) models. By considering ML tasks as items and predictive models as respondents, we can reinterpret the ability of a predictive model in terms of the difficulty and discrimination level of the tasks.

The most recent research can be categorised by how they treated the "items". Martínez-Plumed et al. [26] use IRT to evaluate the predictive performance of ML models on a single classification dataset, treating each instance as an item. They train and test a range of predictive models (i.e., classifiers) on a single dataset and obtain item characteristic curves (ICCs) for each instance. However, the limitation of this framework is its exclusive focus on binary classification tasks and a single dataset. Chen et al. [7] propose a modified IRT model for continuous responses and apply it to multiple classification tasks. The obtained ICCs are not limited to logistic curves, and differently shaped curves can be generated based on the beta distribution, allowing more flexibility when fitting responses for different items. Furthermore, Kandanaarachchi and Smith-Miles [19] treat datasets as respondents, thereby characterising the discrimination and difficulty of the predictive model. They then treat the predictive models as items in an inverted IRT model, to generate the ability trait of datasets, i.e., dataset difficulty.

All of the above IRT models are used to evaluate the predictive performance of ML models, where the response represents the probability of a correct response for the item based on the respondents' ability, whether the items are instances or datasets. However, fairness issues have become increasingly important in real-world applications involving people-related decisions. For example, COMPAS, a decision support model that estimates the risk of a defendant becoming a recidivist, is found to predict a higher risk for black people and a lower risk for white people [5]. Similarly,

Facebook users receive a recommendation prompt when watching a video featuring black people, asking them if they would like to continue watching videos about primates [25]. Another example is Mate AI, an image generator that cannot depict an Asian man and a white woman together [28]. These incidents indicate that datasets or predictive models may become sources of unfairness, leading to serious social problems. We urgently need a fairness evaluation tool to evaluate both datasets and predictive models. Most research typically reports pairwise comparisons between predictive models using various fairness metrics. However, these studies often fail to reveal where and how predictive models falter or to identify the unique strengths and weaknesses of each predictive model. In this paper, we apply IRT to evaluate fairness performance of predictive models and gain meaningful insights into predictive models as well as individuals.

We consider a general scenario for fairness evaluation as shown in Figure 1. A variety of web companies can provide a set of predictive models from AutoML platform for the same task (i.e., classification or regression). The agency has a set of individual observations that are used to evaluate the predictive models. The proposed Fair-IRT framework can be used by the agency to evaluate the fairness performance of the predictive models given by web companies, where the fairness performance is based on a given fairness metric. Note that the Fair-IRT framework is applicable to various fairness metrics and we provide a generality analysis in Appendix A.3.4. In summary, this paper makes the following contributions:

- We propose Fair-IRT, a novel framework to evaluate the fairness performance of individuals as well as predictive models. The parameters learned by Fair-IRT can be used to interpret the ability of predictive models and identify individuals who are treated unfairly. This is the first paper to apply the IRT model in fairness evaluation.
- We propose two ways to disentangle unfairness between individual characteristics and predictive models. The flatness of item characteristic curves (ICCs) is effective for interpretation in the original Fair-IRT setting. Additionally, we introduce a quantitative measure of unfairness by using a Rasch beta IRT model as the backbone of Fair-IRT framework.
- We evaluate the effectiveness of the Fair-IRT framework on two real-world datasets. The experiments demonstrate that Fair-IRT provides comprehensive explanations for fairness evaluation and fosters the development of a more inclusive, equitable, and trustworthy AI.

## 2 Preliminaries

In this section, we present some background of the IRT model and fairness evaluation. We use upper case letters to represent attributes and bold-faced upper case letters to denote the set of attributes. We use bold-faced lower case letters to represent the values of the set of attributes. The values of attributes are represented using lower case letters.

### 2.1 Item Response Theory

In the original context of IRT, *respondents* refer to individuals answering test questions, such as students taking an exam, while *items* refer to the questions or tasks presented to the respondents,

such as specific math problems. We first introduce the logistic IRT model [3] and then briefly discuss the beta IRT model [20], which is the model that we rely on.

We assume a binary response $p_{ij}$ of the $i$-th respondent to the $j$-th item. In the logistic IRT model, the probability of a correct response, i.e., $p_{ij} = 1$, is defined by a logistic function with parameters $\delta_j$ and $a_j$.

The responses are modelled by the Bernoulli distribution with parameter $x_{ij}$ as follows,

$$p_{ij} \sim \text{Bernoulli}(x_{ij}). \quad (1)$$

The logistic IRT framework gives a logistic item characteristic curve (ICC) modelling ability $\theta_i$ to the expected response as follows:

$$\mathbb{E}[p_{ij}|\theta_i, \delta_j, a_j] = x_{ij} = \frac{1}{1 + e^{-a_j(\theta_i - \delta_j)}}. \quad (2)$$

Generally, $\delta_j$ denotes "difficulty", which is the location parameter of the logistic function and can be seen as a measure of item difficulty. $a_j$ indicates "discrimination", which is the steepness of the logistic function at the location point. The above two parameters are relative to items. In contrast, $\theta_i$ is the parameter for the respondent, which is described as the "ability" of the respondent. This parameter is not measured in terms of the number of correct responses but is estimated based on the respondent's responses to discriminating items with different levels of difficulty. Respondents who tend to correctly respond to the most difficult items will be assigned high values of ability.

Then, we introduce the beta IRT model [20], which has been proven to cover more different ICC shapes than the logistic IRT model [7]. It is worth noting that our proposed framework is based on beta IRT model.

In beta IRT, $p_{ij}$ is the observed response of $i$-th respondent to $j$-th item, which is drawn from the Beta distribution,

$$p_{ij} \sim \text{Beta}(\alpha_{ij}, \beta_{ij}),$$
$$\alpha_{ij} = f_\alpha(\theta_i, \delta_j, a_j) = \left(\frac{\theta_i}{\delta_j}\right)^{a_j},$$
$$\beta_{ij} = f_\beta(\theta_i, \delta_j, a_j) = \left(\frac{1 - \theta_i}{1 - \delta_j}\right)^{a_j}, \quad (3)$$

where the parameters $\alpha_{ij}$ and $\beta_{ij}$ are computed by $\theta_i$, $\delta_j$, and $a_j$.

The beta distribution allows us to generate non-logistic ICCs. The ICC is defined as follows,

$$\mathbb{E}[p_{ij}|\theta_i, \delta_j, a_j] = \frac{\alpha_{ij}}{\alpha_{ij} + \beta_{ij}} = \frac{1}{1 + \left(\frac{\delta_j}{1 - \delta_j}\right)^{a_j} \left(\frac{\theta_i}{1 - \theta_i}\right)^{-a_j}}. \quad (4)$$

The ICC describes how an item's performance varies across different levels of a respondent's ability. A typical ICC is an S-shaped curve, indicating the item's difficulty and discrimination. Analysing ICCs helps in assessing the quality of test items and diagnosing the ability characteristics of respondents. Please refer to [7] for further discussion on the advantages of the beta IRT model and its applications.

### 2.2 Fairness Evaluation

We assume a fully supervised learning setting, where the objective is to evaluate fairness for both learned predictive models and individuals. The predictive models are learned over the available

dataset, $\mathcal{D} = \{A, X, Y\}$, where $X$ represents the set of relevant attributes. If we look at the model's prediction $\hat{y} = \hat{Y}(A, X)$, we can assess the fairness of the model's predictions using fairness metrics. These metrics help evaluate whether the model's predictions are influenced by the sensitive attributes or if the predictive model makes fair and unbiased predictions regardless of these attributes.

There are two kinds of fairness metrics, group level and individual level. At the group level, several metrics have been defined such as demographic parity [13], equalised odds [18] and predictive rate parity [41]. However, these group level fairness metrics focus on the population level and do not necessarily mean individual fairness. The fairness metric at the individual level is proposed by Dwork et al. [13] and Louizos et al. [24], but it requires domain knowledge to design a distance function for calculating the similarity between two individuals. For further discussion on the literature regarding fairness evaluation, please refer to Section 6.

## 3 The Proposed Fair-IRT Framework

In this section, we introduce the proposed Fair-IRT framework. We begin by introducing the problem setting including the selected fairness metric. Subsequently, we provide the workflow of the proposed framework.

### 3.1 Problem Setting: Fairness Evaluation using Fair-IRT

We begin by introducing the situational test scores as the fairness metric used in the main text. Please note that the proposed Fair-IRT framework is applicable to other fairness metrics. Further details are provided in Appendix A.2.

Situation tests have been widely employed in the United States as a methodological approach to identify unfairness in recruitment processes [2]. This approach involves controlled experiments designed to analyse employers' hiring decisions based on job applicants' characteristics. Typically, two research assistants with identical qualifications and job-related experience apply for the same position. The key difference between them lies in their sensitive attributes, such as gender, with one applicant being male and the other female. The detection of unfair practices is based on observing discrepancies in favourable decisions between groups differentiated by these sensitive attributes. If the outcomes demonstrate unequal treatment favouring one individual over another, it indicates the presence of unfairness in the hiring process. Additionally, situation tests have been widely recognised as fairness metrics in many research papers [1, 34, 39, 45].

In this paper, the situation test score varies depending on the type of prediction task. The formal definition is as follows:

DEFINITION 1 (SITUATION TEST SCORE (STS)). *Given a predictive model $\hat{Y}^C(\cdot)$ for classification task, the STS is given by:*

$$STS^C = 1 - |P(\hat{Y}^C|A = a, X = x) - P(\hat{Y}^C|A = \bar{a}, X = x)|, \quad (5)$$

*where $P(\cdot)$ denotes the probability estimates for the $\hat{Y}^C(\cdot)$ and $\bar{a}$ denotes the flipped version of the value of the binary sensitive attribute $A$.*

*Given a predictive model $\hat{Y}^R(\cdot)$ for regression task, the STS is given by:*

$$STS^R = 1 - \lambda \left| \frac{\mathbb{E}[\hat{Y}^R|A = a, X = x] - \mathbb{E}[\hat{Y}^R|A = \bar{a}, X = x]}{\mathbb{E}[\hat{Y}^R|A = a, X = x]} \right|, \quad (6)$$

*where $\mathbb{E}[\cdot]$ represents the expected value of the prediction results $\hat{Y}^R$, $\lambda$ is the scaling factor that ensures $STS^R$ falls within the range $[0, 1]$.*

In the following, we ignore the superscript for classification or regression tasks. For a given $\hat{Y}(\cdot)$, STS $> \varepsilon$ indicates that the individual is treated fairly, whereas STS $< \varepsilon$ indicates the opposite. Here, $\varepsilon$ represents the fairness threshold, which is typically determined by domain knowledge. However, in our paper, we set $\varepsilon = 0.5$ as the fairness threshold for simplicity.

As discussed in previous sections, IRT has been applied to evaluate predictive performance in the ML domain. To accommodate different types of tasks, the responses have been redesigned accordingly. For instance, in the context of multi-class classification tasks, the responses represent the probabilities that classifiers assign to the correct class for each instance. In other words, these responses are transformations of accuracy metrics. In our framework, we consider predictive models as *respondents*, individuals as *items*, and the *response* is the result of the selected fairness metric.

More concretely, the predictive models (i.e., $\hat{Y}(\cdot)$) are built using the dataset $\mathcal{D} = \{A, X, Y\}$, which include a binary sensitive attribute $A$, a target attribute $Y$, and a set of relevant attributes $X$. Since the performance of the IRT model depends on the quality of the data [11, 14], we make the following assumption

ASSUMPTION 1. *Given a set of predictive models $\hat{Y}(\cdot)$, these models should exhibit a diversity of fairness performance. Specifically, different predictive models should return different values for the set of individuals when using the same fairness metrics, and these values should be sufficiently sparse.*

In this context, "sparse" refers to the fairness performance should vary significantly across predictive models, where some predictive models demonstrate stronger fairness while others exhibit weaker fairness, ensuring a broad range of fairness outcomes.

It is worth noting that this assumption is both practical and important. We believe that this assumption is easily satisfied in real-world applications. The reason is that different predictive models associate sensitive attributes and target attributes in a black-box manner. The strength of this association cannot be measured under some complex predictive models. Therefore, varying strengths of this association will lead to different fairness performances across different models. The proposed Fair-IRT framework cannot function effectively if all the predictive models achieve the same fairness performance. Evaluating a set of predictive models for fairness is meaningless if all predictive models perform similarly. This underscores the importance of not cherry-picking a set of fair predictive models before implementing Fair-IRT.

### 3.2 The Workflow of Fair-IRT

The backbone of Fair-IRT framework is based on the beta IRT model. It focuses on assessing the fairness performance of a set of predictive models for a set of individuals. In this setting, Fair-IRT can evaluate the ability of predictive models to make fair predictions. Given the

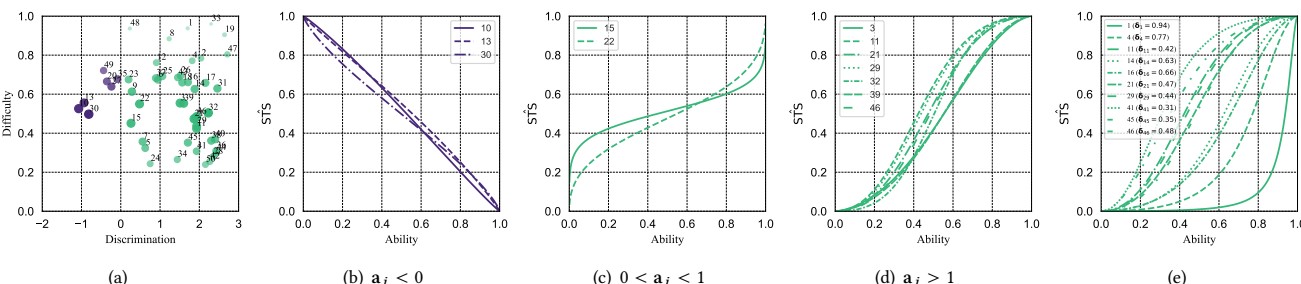

(a)                                         (b) $\mathbf{a}_j < 0$                  (c) $0 < \mathbf{a}_j < 1$                 (d) $\mathbf{a}_j > 1$                              (e)

Figure 2: (a) The scatter plot shows the discrimination parameter $a_j$ and difficulty parameter $\delta_j$ for each individual. The purple points indicate individuals with negative discrimination values (special individuals), while the green points indicate individuals with positive discrimination values (normal individuals). The size of the points increases as $\delta_j$ approaches 0.5 and gradually decreases as $\delta_j$ approaches 0 or 1; (b,c,d) Examples of ICCs generated by Fair-IRT for different values of discrimination and fixed range of difficulty (i.e., $0.4 < \delta_j < 0.6$). Higher discriminations lead to steeper ICCs; (e) Examples of selected ICCs for different values of difficulty and fixed range of discrimination (i.e., $1.7 < a_j < 2$).

dataset $\mathcal{D} = \{A, X, Y\}$ and $M$, where $M$ represents the number of individuals used by the agency to evaluate the predictive models, the workflow is as follows:

i Companies build $N$ predictive models from the dataset $\mathcal{D} = \{A, X, Y\}$. $\hat{Y}_i(\cdot)$ is used to represent the $i$-th predictive model. The set of predictive models $\hat{Y}(\cdot)$ needs to satisfy Assumption 1.

ii The agency evaluates the predictive models provided by web companies using $M$ individuals, as shown in Figure 1. For each predictive model $\hat{Y}_i(\cdot)$, the agency obtains the prediction results for each individual, denoted as $y_{ij}$, where $y_{ij}$ represents the result of the $i$-th predictive model on the $j$-th individual.

iii Depending on the prediction tasks, apply Equation 5 for the classification task and Equation 6 for the regression task. This results in an $N \times M$ matrix containing all STS responses, denoted as $\text{STS}_{ij}$.

iv Apply the beta IRT to this matrix and learn the optimal parameters (i.e., $\delta_j$ and $a_j$ for each individual and $\theta_i$ for each predictive model) that provide the best fit. The learning process is outlined in Algorithm 1 in Appendix A.1.

v Using $\delta_j$, $a_j$ and $\theta_i$ to generate ICCs and provide further insights, including identifying the special individuals that need more attention by the agency (see Figure 2(a)), ranking the predictive models by ability (see Figure 3(a)) and disentangle unfairness between predictive model and individual (see Section 4.3).

## 4 Interpreting Parameters with Simulated Dataset

We use a simulation scenario to analyse and better illustrate our Fair-IRT framework since the ground truth of all parameters is accessible. We have a set of predictive models (i.e., assume 20 predictive models ($N$=20)) that satisfy Assumption 1. In real-world cases, these predictive models are provided by different web companies but are designed for the same tasks. Due to privacy and commercial interests, web companies may not disclose the training and test datasets they used for their predictive models. As an agency, we evaluate the predictive models provided by these web

companies. We can access a set of individuals used for evaluation (i.e., assume 50 individuals ($M$=50)).

### 4.1 Individual Parameters: Discrimination and Difficulty

The Fair-IRT framework comprises two parameters per individual: discrimination and difficulty. In this case, 50 individual ICCs are derived (one per individual), and 20 values of ability for the set of predictive models are estimated. Figure 2(a) shows the discrimination and difficulty values for each individual.

*4.1.1 Discrimination.* The discrimination parameter measures an individual's capability to differentiate between predictive models. Therefore, when applying Fair-IRT, the discrimination parameter of an individual can indicate if the individual is a special case. Of the 50 individuals, 43 had positive discrimination values (i.e., the green points in Figure 2(a)), and the selected ICCs are shown in Figure 2(c) and Figure 2(d). These cases are normal, as an increase in the fair ability of predictive models corresponds to an increase in their STS.

However, negative discrimination values are observed in 7 individuals (i.e., the purple points in Figure 2(a)). We plot the selected ICCs in Figure 2(b). Since the discrimination is negative, it indicates that these individuals are most frequently treated fairly by the most unfair predictive models and unfairly by fair predictive models. Such cases are typically referred to as special individuals and should be identified by the agency for further analysis.

In summary, Figures 2(b), 2(c) and 2(d) show examples of Item Characteristic Curves (ICCs) generated by Fair-IRT framework for different values of the discrimination parameter $a_j$:

- $a_j < 0$: the ICC shows a decreasing trend.
- $0 < a_j < 1$: the ICC demonstrates an anti-sigmoidal behaviour, indicating a slower increase followed by a rapid increase.
- $a_j > 1$: the ICC exhibits an "S"-shaped (sigmoid) curve,

Notably, Fair-IRT framework allows for negative discrimination values, indicating individuals who are special and require further analysis.



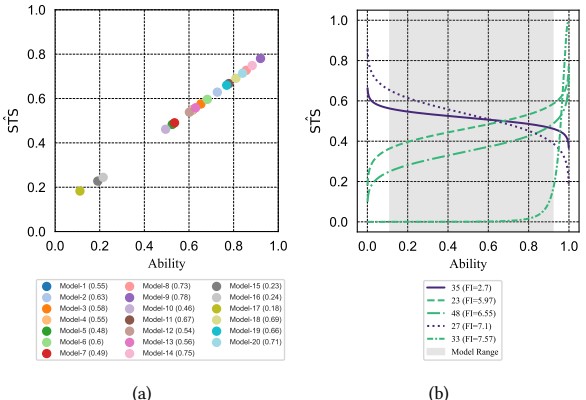

(a)       (b)

**Figure 3: (a) The scatter plot shows the $\hat{\text{STS}}$ and ability parameter $\theta_i$ for each predictive model; (b) Examples of selected individuals with flat ICCs. The shaded area indicates the ability range of the 20 selected predictive models.**

*4.1.2 Difficulty.* The parameter difficulty ($\delta_j$) provides a straightforward yet powerful measure of the likelihood that an individual will be treated fairly or unfairly by predictive models. The range of difficulty values is from 0 to 1, where:

- Individuals with $\delta_j$ close to 1: These individuals are unfairly treated by almost all predictive models, indicating a high difficulty in achieving fair prediction.
- Individuals with $\delta_j$ close to 0: These individuals are consistently fairly treated by all predictive models, indicating a low difficulty in achieving fair prediction.

In Figure 2(e), individual "1" is more likely to be unfairly treated by almost all predictive models, as evidenced by a difficulty value close to 1. We assert that the unfairness experienced by this individual arises from their inherent characteristics, resulting in consistently lower STS regardless of the fairness ability of the predictive models. This suggests that the difficulty parameter effectively captures intrinsic factors contributing to the likelihood of fair prediction.

The difficulty parameter helps identify individuals who are persistently vulnerable to unfair prediction. By flagging these individuals, further investigation can be conducted to understand and address the specific factors contributing to their unfair prediction. Recognising individuals with high difficulty values can inform targeted interventions. For example, if certain individuals consistently exhibit high difficulty values, it may indicate underlying systemic biases that need to be addressed through policy changes or tailored fairness initiatives.

## 4.2 Predictive Model Parameter: Ability

As we mentioned, Fair-IRT framework offers dual analysis directions, providing valuable information about both individuals and predictive models. The Fair-IRT framework estimates an ability value for each predictive model, denoted as $\theta_i$. The results for ability and $\hat{\text{STS}}_i$ are shown in Figure 3(a). $\hat{\text{STS}}_i$ denotes the estimated

STS for each predictive model, calculated using the following formula:

$$\hat{\text{STS}}_i = \frac{1}{N} \sum_j^N \hat{\text{STS}}_{ij}. \tag{7}$$

The scatter plot in Figure 3(a) indicates a positive correlation between the ability parameter $\theta_i$ and the estimated $\hat{\text{STS}}_i$. This suggests that predictive models with higher ability are generally more fair in their predictions across individuals. The scatter plot also allows for the identification of outliers. The predictive models that significantly deviate from the general trend can be flagged for further investigation. For instance, a predictive model with high ability but low $\hat{\text{STS}}$, or vice versa, may indicate potential areas of bias or performance issues. While this situation does not occur in our examples, it remains a possibility in other scenarios.

## 4.3 Disentangle Unfairness between Predictive Model and Individual

We first attempt to disentangle the unfairness between the predictive model and the individual under the backbone of the beta IRT model. We find this approach limited, as it introduces an interaction component involving the combination of the predictive model and the individual. However, by using ICCs for each individual, we can identify certain patterns. An ICC with a flat curve within a certain ability range can be recognised as indicating unfairness originating from individuals. For example, individual "1" from Figure 2(e) and individual "15" from Figure 2(c) both show a flat curve within the ability range of $(0.4, 0.6)$. The underlying reasons for these flat curves differ: individual "1" has a discrimination parameter close to 0, while individual "15" has a difficulty parameter close to 1. Thus, we should consider parameters for individuals simultaneously.

We now formally define flatness in ICCs mathematically. The $\hat{\text{STS}}_j$ can be considered a function of $\theta$ for a given $j$-th individual and is given as follows:

$$\hat{\text{STS}}_j = f(\theta_i) = \frac{1}{1 + \left(\frac{\delta_j}{1-\delta_j}\right)^{a_j} \left(\frac{\theta_i}{1-\theta_i}\right)^{-a_j}}. \tag{8}$$

To find the derivative of $f(\theta_i)$, follow these steps:
Step 1. Define the intermediate function $h(\theta_i)$:

$$h(\theta_i) = 1 + \left(\frac{\delta_j}{1-\delta_j}\right)^{a_j} \left(\frac{\theta_i}{1-\theta_i}\right)^{-a_j}.$$

Step 2. Compute the derivative of $h(\theta_i)$:

$$\frac{d}{d\theta}\left(\frac{\theta_i}{1-\theta_i}\right) = \frac{(1-\theta_i) - \theta_i(-1)}{(1-\theta_i)^2} = \frac{1}{(1-\theta_i)^2}.$$

Let $u = \frac{\theta_i}{1-\theta_i}$, then $h(\theta_i) = 1 + \left(\frac{\delta_j}{1-\delta_j}\right)^{a_j} u^{-a_j}$. The derivative of $u^{-a_j}$ with respect to $\theta$ is:

$$\frac{d}{d\theta}(u^{-a_j}) = -a_j u^{-a_j-1} \cdot \frac{1}{(1-\theta_i)^2}.$$

Thus,

$$h'(\theta_i) = \left(\frac{\delta_j}{1-\delta_j}\right)^{a_j} \left(-a_j \left(\frac{\theta_i}{1-\theta_i}\right)^{-a_j-1} \cdot \frac{1}{(1-\theta_i)^2}\right).$$

Step 3. Finally, compute the derivative of $f(\theta_i)$:

$$f'(\theta_i) = -\frac{h'(\theta_i)}{(h(\theta_i))^2}.$$

Substitute $h(\theta_i)$ and $h'(\theta_i)$:

$$f'(\theta_i) = -\frac{\left(\frac{\delta_j}{1-\delta_j}\right)^{a_j}\left(-a_j\left(\frac{\theta_i}{1-\theta_i}\right)^{-a_j-1}\cdot\frac{1}{(1-\theta_i)^2}\right)}{\left(1+\left(\frac{\delta_j}{1-\delta_j}\right)^{a_j}\left(\frac{\theta_i}{1-\theta_i}\right)^{-a_j}\right)^2}.$$

Simplify to get the final derivative of $f(\theta_i)$:

$$f'(\theta_i) = \frac{a_j\left(\frac{\delta_j}{1-\delta_j}\right)^{a_j}\left(\frac{\theta_i}{1-\theta_i}\right)^{-a_j-1}}{\left(1+\left(\frac{\delta_j}{1-\delta_j}\right)^{a_j}\left(\frac{\theta_i}{1-\theta_i}\right)^{-a_j}\right)^2}\cdot\frac{1}{(1-\theta_i)^2}. \quad (9)$$

Given an individual $j$, the Flatness Indicator (FI) for ICC is defined as follows,

$$\text{FI}_j = \sum_i^M |f'(\theta_i)|. \quad (10)$$

To demonstrate the effectiveness of the FI, we select a range of predictive models and plot the ICCs for individuals with the five smallest FI. It is important to note that the number of selections may vary depending on the evaluation set and task. Figure 3(b) illustrates the ICCs for the selected individuals and predictive models. The selected individuals exhibit very low $\text{FI}_j$ values, which indicates a flat ICC. In this scenario, for these individuals, the unfairness stems from the individuals themselves, as the ability to increase $\hat{\text{STS}}$ remains at a very similar level.

Furthermore, we apply a specialised way where the backbone of Fair-IRT framework is based on the Rasch beta IRT model. This way focuses on quantitatively disentangling the unfairness between individuals and predictive models. We keep all other steps the same as in the general setting but set the parameter $a_j = 1$ as a constant. The ICCs are given as follows,

$$\hat{\text{STS}}_{ij} = \frac{1}{1+\left(\frac{\delta_j}{1-\delta_j}\right)\left(\frac{1-\theta_i}{\theta_i}\right)}. \quad (11)$$

Now, we do some transformations on the above formula,

$$\implies 1+\left(\frac{\delta_j}{1-\delta_j}\right)\left(\frac{1-\theta_i}{\theta_i}\right) = \frac{1}{\hat{\text{STS}}_{ij}}$$

$$\implies \left(\frac{\delta_j}{1-\delta_j}\right)\left(\frac{1-\theta_i}{\theta_i}\right) = \frac{1-\hat{\text{STS}}_{ij}}{\hat{\text{STS}}_{ij}}$$

$$\implies \log\left(\frac{\delta_j}{1-\delta_j}\right)+\log\left(\frac{1-\theta_i}{\theta_i}\right) = \log(1-\hat{\text{STS}}_{ij})-\log\hat{\text{STS}}_{ij}.$$

Let $\Delta_j = \frac{\delta_j}{1-\delta_j}$ and $\Theta_i = \frac{1-\theta_i}{\theta_i}$, then we have,

$$\log\Delta_j + \log\Theta_i = \log(1-\hat{\text{STS}}_{ij})-\log\hat{\text{STS}}_{ij}. \quad (12)$$

Here, $\Delta_j$ is the quantity of unfairness from the individual and $\Theta_i$ is the quantity of unfairness from the predictive model.

The Equation 12 can be rewritten as follows,

$$g(\hat{\text{STS}}_{ij}) = \log\Delta_j + \log\Theta_i, \quad (13)$$

where $g(\hat{\text{STS}}_{ij}) = \log(1-\hat{\text{STS}}_{ij})-\log\hat{\text{STS}}_{ij}$.

To conduct a straightforward analysis, we define predictions with $\hat{\text{STS}}_{ij} < 0.5$ as unfair, such that $g(\hat{\text{STS}}_{ij}) > 0$. We maintain the same simulation process and focus on individual "1" with the same predictive models as shown in Figure 3(b). We observe that $g(\hat{\text{STS}}) = 2.83$ for individual "1" on predictive model "5," indicating an unfair prediction. The values $\log\Delta = 3.07$ and $\log\Theta = -0.24$ represent the quantity of unfairness from the individual and the predictive model, respectively. This indicates that the unfairness primarily originates from the individual characteristics, consistent with our previous discussion.

In summary, both ways can provide insights into disentangling unfairness between individual and predictive model. However, the quantitative way is more suitable for less complex situations, since the Rasch beta IRT is weaker in fitting power than the original beta IRT. We suggest using the quantitative way to supplement the explanation.

## 5 Experiments with two Real-world Datasets

In this section, we apply the Fair-IRT framework to two real-world datasets and focus on different types of tasks. We simulate the generation of a set of non-fairness-aware predictive models. To achieve this, we emulate the web company's mechanism for generating non-fairness-aware predictive models by using the AutoML platform, which produces a set of highly accurate predictive models across different types. All predictive models are implemented using the H2O package [22], which includes various categories such as the generalised linear model (DLM), deep learning model (DP), tree-based model (XRT or DRT), gradient boosting model (GBM), and stacked ensemble model (SE). We distinguish the predictive models by their shorthand model names and numbers. The source code for the predictive models is available via the link provided in the abstract. We employ 10-fold cross-validation to train and evaluate these predictive models.

We note that the proposed Fair-IRT framework is not restricted by the choice of sensitive attributes or fairness metrics. Due to space limitations, we provide additional experiments for different sensitive attributes and fairness metrics in Appendix A.3.3 and Appendix A.3.4, respectively. These aim to demonstrate the generalisation ability of the proposed Fair-IRT framework.

### 5.1 Adult

The Adult dataset comes from the UCI repository [12] and contains 14 attributes including race, age, education information, marital information as well as capital gain and loss for 48,842 individuals. We pre-process the dataset by deleting missing information and encoding discrete attributes. The downstream tasks' goal is to predict whether the individual's income is above $50,000, which belongs to the classification task. We set *sex* as sensitive attribute and all the other attributes as non-sensitive attributes. We randomly select 1,000 individuals as the evaluation set.

We simulate 24 predictive models for the Adult dataset. The predictive performance is measured by the area under the curve (AUC) and the results are shown in Table 1. We use Equation 5 as fairness metrics since it is a classification task. We set STS > 0.5 as the threshold for considering the individual is treated fairly by the predictive model. We plot the ICCs for five individuals with

**Table 1: The predictive performance (AUC), ability parameter $\theta_i$, and estimated $\hat{STS}_i$ for 24 predictive models on the Adult dataset are presented. Note that $\hat{STS}_i$ represents the estimated STS for each predictive model, computed using Equation 7.**

| Model | AUC | Ability | $\hat{STS}_i$ | Model | AUC | Ability | $\hat{STS}_i$ |
|---|---|---|---|---|---|---|---|
| SE_1 | 0.929 | 0.883 | 0.793 | SE_3 | 0.851 | 0.810 | 0.694 |
| GBM_1 | 0.928 | 0.865 | 0.764 | GBM_3 | 0.839 | 0.845 | 0.732 |
| XRT_1 | 0.920 | 0.935 | 0.864 | DL_3 | 0.837 | 0.541 | 0.451 |
| DRF_1 | 0.919 | 0.831 | 0.694 | XRT_3 | 0.831 | 0.838 | 0.715 |
| DL_1 | 0.914 | 0.777 | 0.675 | DRF_3 | 0.822 | 0.613 | 0.492 |
| GLM_1 | 0.913 | 0.473 | 0.415 | GLM_3 | 0.822 | 0.436 | 0.381 |
| SE_2 | 0.902 | 0.851 | 0.753 | SE_4 | 0.819 | 0.851 | 0.719 |
| GBM_2 | 0.902 | 0.810 | 0.701 | GBM_4 | 0.816 | 0.912 | 0.771 |
| XRT_2 | 0.900 | 0.945 | 0.864 | XRT_4 | 0.816 | 0.821 | 0.690 |
| DRF_2 | 0.873 | 0.791 | 0.654 | DRF_4 | 0.811 | 0.546 | 0.443 |
| DL_2 | 0.855 | 0.715 | 0.625 | DL_4 | 0.765 | 0.618 | 0.551 |
| GLM_2 | 0.851 | 0.482 | 0.437 | GLM_4 | 0.745 | 0.435 | 0.416 |

**Table 2: The quantitative way of disentangling Individual "343" from the Adult dataset. ✓ indicates that the individual is treated fairly under the selected predictive model, while ✗ indicates the opposite.**

| | | Individual 343 (log $\Delta_j$ = 1.236) | | | | |
|---|---|---|---|---|---|---|---|
| Model | $\log\Theta_i$ | $g(\hat{STS}_{ij})$ | | Model | $\log\Theta_i$ | $g(\hat{STS}_{ij})$ | |
| SE_1 | -2.163 | -0.928 | ✓ | SE_3 | -1.541 | -0.305 | ✓ |
| GBM_1 | -1.977 | -0.742 | ✓ | GBM_3 | -1.769 | -0.534 | ✓ |
| XRT_1 | -2.863 | -1.627 | ✓ | DL_3 | -0.273 | 0.962 | ✗ |
| DRF_1 | -1.599 | -0.364 | ✓ | XRT_3 | -1.705 | -0.469 | ✓ |
| DL_1 | -1.398 | -0.162 | ✓ | DRF_3 | -0.467 | 0.768 | ✗ |
| GLM_1 | -0.049 | 1.187 | ✗ | GLM_3 | 0.200 | 1.435 | ✗ |
| SE_2 | -1.863 | -0.628 | ✓ | SE_4 | -1.776 | -0.540 | ✓ |
| GBM_2 | -1.549 | -0.314 | ✓ | GBM_4 | -2.286 | -1.050 | ✓ |
| XRT_2 | -2.922 | -1.687 | ✓ | XRT_4 | -1.545 | -0.310 | ✓ |
| DRF_2 | -1.327 | -0.091 | ✓ | DRF_4 | -0.236 | 1.000 | ✗ |
| DL_2 | -1.131 | 0.105 | ✗ | DL_4 | -0.748 | 0.488 | ✗ |
| GLM_2 | -0.173 | 1.063 | ✗ | GLM_4 | -0.020 | 1.216 | ✗ |

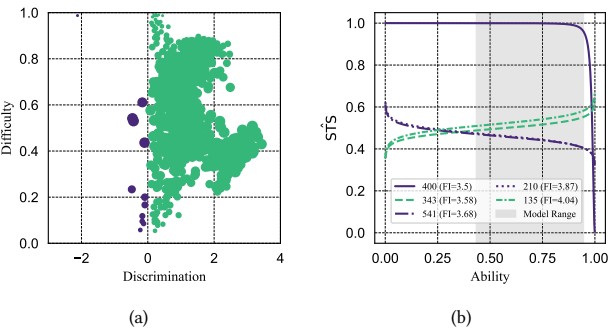

(a)                    (b)

**Figure 4: The plots for the Adult dataset: (a) The scatter plot shows the discrimination parameter $a_j$ and the difficulty parameter $\delta_j$ for each individual; (b) Examples of selected individuals with flat ICCs. The shaded area indicates the ability range of the 24 selected predictive models.**

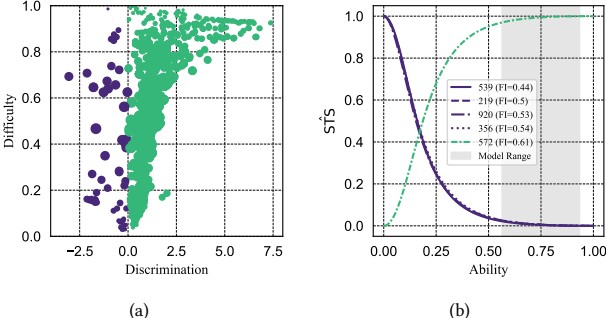

(a)                    (b)

**Figure 5: The plots for the Law School dataset: (a) The scatter plot shows the discrimination parameter $a_j$ and the difficulty parameter $\delta_j$ for each individual; (b) Examples of selected individuals with flat ICCs. The shaded area indicates the ability range of the 15 selected predictive models.**

the smallest flatness indicators. After using Fair-IRT, we have the following observations,

- Table 1 shows the predictive performance (AUC), ability parameter $\theta_i$, and estimated $\hat{STS}_i$ for 24 predictive models on the Adult dataset. We note that XRT_2 and XRT_1 achieve the highest fairness performance. However, these two predictive models are not the best model in predictive performance.
- Figure 4(a) is the scatter plot of the discrimination parameter $a_j$ and difficulty parameter $\delta_j$ for each individual in evaluation set. The purple dot denotes an individual identified as a special case, where the value of the discrimination parameter is negative. The agency should flag this individual for further analysis, as they are treated more fairly by a lower-ability predictive model.
- Figure 4(b) presents the ICCs for individuals with the five smallest FI. Table 2 shows the results of quantitatively disentangling individual "343". Under the predictive model "GLM_1", we find that the individual component contributes significantly more

than the predictive model component, whereas under the predictive model "DL_2", the predictive model component contributes more. The agency should take further action on these individuals with unfair predictions, which may include but is not limited to, adjusting the predictive model. The results for additional selected individuals are provided in Appendix A.3.1.

## 5.2 Law School

The law school dataset comes from a survey [35] of admissions information from 163 law schools in the United States. It contains information of 21,790 law students, including their entrance exam scores (LSAT), their grade point average (GPA) collected prior to law school, and their first-year average grade (FYA). The school expects to predict if the applicants will have a high FYA. Gender is the sensitive attribute in this dataset, and the school also wants to

ensure that predictions are not affected by the sensitive attribute. We randomly select 1, 000 individuals as the evaluation set.

We simulate 15 predictive models for the law school dataset, measuring predictive performance using root mean square error (RMSE). The results are presented in Table 4 in Appendix A.3.2. Since this is a regression task, we use Equation 6 as the fairness metric, setting STS > 0.5 as the threshold for considering an individual to be treated fairly by the predictive model. Using Fair-IRT, we identify some special individuals with negative discrimination, as shown in Figure 5(a). Figure 5(b) highlights ICCs for five individuals with the smallest FI. Individual "572" exhibits a different pattern. Individual "572" has a flat ICC with a high value of STS, as shown in Figure 5(b). Table 5 in Appendix A.3.2 presents the results of the quantitative analysis for disentangling individual "572". This suggests that this individual is consistently privileged and treated fairly, regardless of the predictive model's ability.

## 6 Related Works

**Item Response Theory and its Application.** Item Response Theory (IRT) describes a group of models that explore how latent traits (e.g., intelligence) influence observed responses (e.g. assessment score) [11, 14, 17]. Specifically, IRT models the variables that cannot be directly observed, such as language skills, attitudes towards different races, or susceptibility to stress, which are considered latent traits. These latent traits can be used to explain why people respond the way when they do questionnaires or surveys. By linking items (i.e., questionnaires or surveys) to respondents' latent traits, IRT effectively provides a way to compare them. This theory has been widely applied in psychometrics [9] and educational testing [40]. There are a variety of models developed in IRT for different types of responses. For example, the logistic IRT model is designed for binary responses, in which the responses are either correct or incorrect; [32] propose the multi-response ordinal models for polytomous data; the Continuous Response Model (CRM) as an extension of polytomous IRT is designed for continuous response [33].

In recent decades, decision-making models have been applied in many fields. People realise that some tasks are more difficult than others, and some predictive models are more capable than others. Is it a monotonic one, i.e., better techniques usually get better results on more difficult problems and usually solve the easier ones? Interestingly, all of these issues have been addressed in the past by IRT, yet in very different contexts. Martínez-Plumed et al. [26] use IRT to evaluate the predictive performance of predictive models on a signal classification dataset; Chen et al. [7] propose a modified IRT model for continuous responses and use it to evaluate multiple classification tasks; Kandanaarachchi and Smith-Miles [19] generate an inverted version of IRT model and evaluate a set of models across a repository of datasets. It is worth noting that all the above IRT frameworks are used to evaluate the predictive performance of the predictive models.

**Fairness Evaluation.** The machine learning literature has increasingly focused on evaluating how models can protect marginalised populations from unfair treatment. An important direction is how to quantify fairness, i.e., the fairness metrics. By using these fairness metrics, we can rank models according to their overall results or even do pairwise comparisons and show that method A is more fair

than B. In the statistical framework, Demographic parity is defined by Zemel et al. [42], which is used to measure group-level fairness. Other similar metrics include equalised odds [18], predictive rate parity [41]. Dwork et al. [13] propose a measurement to quantify individual-level fairness, i.e., similar individuals should have similar treatments, and they use distance functions to measure how similar between individuals. Apart from the statistical framework, some metrics are developed under the causal framework, which focuses on causal relationships rather than associate relationships. The (conditional) average causal effect is used to quantify fairness between groups [23]; Natural direct and natural indirect effects are used to quantify specific fairness [29, 36, 37, 43, 46]; When unfair causal paths are identified by domain knowledge, Chiappa [8] used the path-specific causal effects to quantify fairness on approved paths; Kusner et al. [21] introduce the definition of counterfactual fairness which can be used to answer what-if questions in fairness machine learning [16] and develop counterfactual fair predictive models [38]. For more related works, please refer to the literature review [6, 10, 15, 27, 31, 44]. However, the above literature contributes to developing more specific rulers for evaluating fairness. We still do not know how the overall fairness performance for a collection of benchmark predictive models or specific individuals is distributed.

Our paper is a novel lens of fairness evaluation by bringing the IRT model. We can evaluate the fairness performance of a set of predictive models on different individuals and obtain the latent fairness ability of the predictive models. Through the flatness of ICCs, we can also disentangle the unfairness between individuals and predictive models. To the best of our knowledge, this is the first paper to use the IRT model to evaluate fairness performance for both predictive models and individuals.

## 7 Conclusion

**Summary of Contributions.** In this paper, we introduce Fair-IRT, a novel framework based on beta IRT, to evaluate the fairness performance of both predictive models and individuals. This is the first paper to apply the IRT model in fairness evaluation. The parameters learned by Fair-IRT can be used to interpret the ability of predictive models and identify individuals who are treated unfairly. Furthermore, we propose two ways to disentangle unfairness between individuals and predictive models. The flatness of item characteristic curves is proposed for the original setting of Fair-IRT and is effective for interpretation. A quantitative way to measure the composition of unfairness is proposed by replacing the backbone with the Rasch beta IRT. Our experimental evaluation of real-world datasets demonstrates the effectiveness of the Fair-IRT framework. The results show that the proposed Fair-IRT provides comprehensive explanations for fairness evaluation and promotes the development of more inclusive, equitable, and trustworthy AI.

**Limitations & Future Works.** The proposed Fair-IRT framework currently operates on a single fairness metric and sensitive attribute. In the future, we plan to explore a high-dimensional IRT framework capable of addressing both utility and fairness metrics simultaneously. We also intend to apply the Fair-IRT framework to fairness-aware predictive models to compare their fairness at the application level.

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

# A Appendix

This is the Appendix for "Fairness Evaluation with Item Response Theory".

## A.1 Learning Algorithm for Fair-IRT

The learning process is outlined in Algorithm 1.

---
**Algorithm 1**

---
**Inputs:**
    The matrix $\text{STS}_{ij}$, including situation test score of the $j$-th individual on the $i$-th predictive model.
    *epochs*: Number of training epochs
**Outputs:**
    $\theta$: Estimated abilities of predictive models
    $\delta$: Estimated difficulties of individuals
    $a$: Estimated discriminations of individuals
Initialise parameters: $\theta$, $\delta$ and $a$
**for** *epochs* = 1 to *epochs* **do**
    $\hat{\text{STS}}_{ij} \leftarrow$ 3-parameter beta IRT$(\theta, \delta, a)$
    $current\_loss \leftarrow loss(\text{STS}_{ij}, \hat{\text{STS}}_{ij})$
    $gradients \leftarrow gradient(current\_loss, [\theta, \delta, a])$
**end for**
Extract parameters:
    $\theta \leftarrow sigmoid(\theta)$, $\delta \leftarrow sigmoid(\delta)$ and $a \leftarrow a$
Return $(\theta, \delta, a)$

---

## A.2 Supplementary Fairness Metrics

We first introduce additional fairness metrics and explain why they are not selected in the main text. Additionally, we adapt certain group level fairness metrics to make them suitable for use within our framework.

**DEFINITION 2 (INDIVIDUAL FAIRNESS (IF) [13, 24]).** *A predictive model is fair if it gives similar predictions to similar individuals. Formally, given a distance function $d(\cdot, \cdot)$, if individuals $j$ and $k$ are similar under this distance function (i.e., $d(j, k)$ is small) then their predictions should be similar:*

$$\hat{Y}(A_j, X_j) \approx \hat{Y}(A_k, X_k). \quad (14)$$

We note that the distance function $d(\cdot, \cdot)$ must be carefully chosen, requiring an in depth understanding of the domain knowledge.

Another individual level fairness metric is counterfactual fairness, which belongs to the causal framework and is defined as follows:

**DEFINITION 3 (COUNTERFACTUAL FAIRNESS [21]).** *Prediction model $\widehat{Y}(\cdot)$ is counterfactually fair if under any context $X = x$ and $A = a$,*

$$P(\widehat{Y}_{A \leftarrow a}(U) = y \mid X = x, A = a) = \\ P(\widehat{Y}_{A \leftarrow \bar{a}}(U) = y \mid X = x, A = a), \quad (15)$$

*for all $y$ and any value $\bar{a}$ attainable by $A$. $U$ is a set of the background attributes, which are the factors not caused by any attributes in the set $\{A, X\}$.*

The counterfactual is modelled as the solution for $Y$ for a given $U = u$, where the equations for $A$ are replaced with $A = a$. We denote it by $Y_{A \leftarrow a}(U)$. However, the calculation process for counterfactual fairness is difficult to satisfy in real-world applications. It

requires complex steps [30] and strong assumptions, i.e., the prior knowledge of the structural equation model [4].

It is important to note that the fairness metrics suitable for the proposed Fair-IRT framework should link the prediction results with the sensitive attribute, rather than focusing solely on the target variable. Details of the selected group fairness metrics are provided as follows,

- dp (Demographic Parity or Statistical Parity) [13]. A predictive model satisfies demographic parity if the prediction $\hat{y}$ is independent of the sensitive attribute $A$, i.e., $P(\hat{Y}|A = 0) = P(\hat{Y}|A = 1)$.
- eopp (Equality of Opportunity) [18]. A predictive model satisfies equalised opportunity if the prediction $\hat{y}$ is independent of the sensitive attribute $A$ when the label $Y = 1$, i.e., $P(\hat{Y}|A = 0, Y = 1) = P(\hat{Y}|A = 0, Y = 1)$.
- eodd (Equalised Odds) [18]. A predictive model satisfies equalised odds if the prediction $\hat{y}$ is independent of the sensitive attribute $A$ conditioned on the label $Y$, i.e., $P(\hat{Y}|A = 0, Y = y) = P(\hat{Y}|A = 1, Y = y)$, where $y \in \{0, 1\}$.

The above fairness metrics can be extended to the individual level by adding conditions on the set of attributes associated with individual. The details are provided as follow,

- dp: $P(\hat{Y}|A = 0, X = x) = P(\hat{Y}|A = 1, X = x)$.
- eopp: $P(\hat{Y}|A = 0, Y = 1, X = x) = P(\hat{Y}|A = 0, Y = 1, X = x)$.
- eodd: $P(\hat{Y}|A = 0, Y = y, X = x) = P(\hat{Y}|A = 1, Y = y, X = x)$, where $y \in \{0, 1\}$.

We note that demographic parity (dp) is a restricted version of the situation test score (the fairness metric used in the main text). This is because dp requires the probability to remain equal when the sensitive attribute is flipped. In contrast, the situation test score allows for a difference (i.e., a threshold $\varepsilon$) that can be adjusted by the end user.

We can define the equalised score using the aforementioned eopp and eodd metrics for the proposed Fair-IRT framework. The formal definition is as follows:

**DEFINITION 4 (EQUALISED SCORE (ES)).** *Given a predictive model $\hat{Y}^C(\cdot)$ for classification task, the ES is given by:*

$$ES^C = 1 - |P(\hat{Y}^C|A = a, Y = y, X = x) - P(\hat{Y}^C|A = \bar{a}, Y = y, X = x)|,$$

*where $P(\cdot)$ denotes the probability estimates for the $\hat{Y}^C(\cdot)$, $\bar{a}$ denotes the flipped version of the value of the binary sensitive attribute $A$.*

*Given a predictive model $\hat{Y}^R(\cdot)$ for regression task, the ES is given by:*

$$ES^R = 1 - \lambda \left| \frac{\mathbb{E}[\hat{Y}^R|A = a, Y = y, X = x] - \mathbb{E}[\hat{Y}^R|A = \bar{a}, Y = y, X = x]}{\mathbb{E}[\hat{Y}^R|A = a, Y = y, X = x]} \right|,$$

*where $\mathbb{E}[\cdot]$ represents the expected value of the prediction results $\hat{Y}^R$, $\lambda$ is the scaling factor that ensures $ES^R$ falls within the range $[0, 1]$.*

## A.3 Supplementary Experimental Results

*A.3.1 Experimental results for additional selected individuals on the Adult Dataset.* In this section, we provide detailed results for selected individuals from the Adult dataset in Table 3. The selected individuals are those with the five smallest flatness indicators. Our aim is to quantitatively disentangle unfairness between individuals and predictive models.

**Table 3: The quantitative way of disentangling selected individuals from the Adult dataset.** ✓ indicates that the individual is treated fairly under the selected predictive model, while ✗ indicates the opposite.

| Model | $\log \Theta_i$ | Individual 400 ($\log \Delta_j$=-1.333) $g(\hat{STS}_{ij})$ | | Individual 343 ($\log \Delta_j$=1.236) $g(\hat{STS}_{ij})$ | | Individual 541 ($\log \Delta_j$=1.784) $g(\hat{STS}_{ij})$ | | Individual 210 ($\log \Delta_j$=1.793) $g(\hat{STS}_{ij})$ | | Individual 135 ($\log \Delta_j$=0.869) $g(\hat{STS}_{ij})$ | |
|---|---|---|---|---|---|---|---|---|---|---|---|
| SE_1 | -2.163 | -3.496 | ✓ | -0.928 | ✓ | -0.379 | ✓ | -0.370 | ✓ | -1.294 | ✓ |
| GBM_1 | -1.977 | -3.310 | ✓ | -0.742 | ✓ | -0.193 | ✓ | -0.184 | ✓ | -1.108 | ✓ |
| XRT_1 | -2.863 | -4.196 | ✓ | -1.627 | ✓ | -1.079 | ✓ | -1.070 | ✓ | -1.993 | ✓ |
| DRF_1 | -1.599 | -2.932 | ✓ | -0.364 | ✓ | 0.185 | ✗ | 0.194 | ✗ | -0.730 | ✓ |
| DL_1 | -1.398 | -2.731 | ✓ | -0.162 | ✓ | 0.386 | ✗ | 0.395 | ✗ | -0.528 | ✓ |
| GLM_1 | -0.049 | -1.382 | ✓ | 1.187 | ✗ | 1.735 | ✗ | 1.744 | ✗ | 0.821 | ✗ |
| SE_2 | -1.863 | -3.196 | ✓ | -0.628 | ✓ | -0.079 | ✓ | -0.070 | ✓ | -0.994 | ✓ |
| GBM_2 | -1.549 | -2.882 | ✓ | -0.314 | ✓ | 0.235 | ✗ | 0.244 | ✗ | -0.680 | ✓ |
| XRT_2 | -2.922 | -4.255 | ✓ | -1.687 | ✓ | -1.138 | ✓ | -1.129 | ✓ | -2.053 | ✓ |
| DRF_2 | -1.327 | -2.660 | ✓ | -0.091 | ✓ | 0.457 | ✗ | 0.466 | ✗ | -0.458 | ✓ |
| DL_2 | -1.131 | -2.464 | ✓ | 0.105 | ✗ | 0.653 | ✗ | 0.662 | ✗ | -0.262 | ✓ |
| GLM_2 | -0.173 | -1.506 | ✓ | 1.063 | ✗ | 1.611 | ✗ | 1.620 | ✗ | 0.697 | ✗ |
| SE_3 | -1.541 | -2.874 | ✓ | -0.305 | ✓ | 0.243 | ✗ | 0.252 | ✗ | -0.671 | ✓ |
| GBM_3 | -1.769 | -3.102 | ✓ | -0.534 | ✓ | 0.015 | ✗ | 0.024 | ✗ | -0.900 | ✓ |
| DL_3 | -0.273 | -1.607 | ✓ | 0.962 | ✗ | 1.510 | ✗ | 1.520 | ✗ | 0.596 | ✗ |
| XRT_3 | -1.705 | -3.038 | ✓ | -0.469 | ✓ | 0.079 | ✗ | 0.088 | ✗ | -0.836 | ✓ |
| DRF_3 | -0.467 | -1.800 | ✓ | 0.768 | ✗ | 1.317 | ✗ | 1.326 | ✗ | 0.402 | ✗ |
| GLM_3 | 0.200 | -1.133 | ✓ | 1.435 | ✗ | 1.984 | ✗ | 1.993 | ✗ | 1.069 | ✗ |
| SE_4 | -1.776 | -3.109 | ✓ | -0.540 | ✓ | 0.008 | ✗ | 0.017 | ✗ | -0.906 | ✓ |
| GBM_4 | -2.286 | -3.619 | ✓ | -1.050 | ✓ | -0.502 | ✓ | -0.493 | ✓ | -1.416 | ✓ |
| XRT_4 | -1.545 | -2.879 | ✓ | -0.310 | ✓ | 0.238 | ✗ | 0.247 | ✗ | -0.676 | ✓ |
| DRF_4 | -0.236 | -1.569 | ✓ | 1.000 | ✗ | 1.548 | ✗ | 1.557 | ✗ | 0.633 | ✗ |
| DL_4 | -0.748 | -2.081 | ✓ | 0.488 | ✗ | 1.036 | ✗ | 1.045 | ✗ | 0.121 | ✗ |
| GLM_4 | -0.020 | -1.353 | ✓ | 1.216 | ✗ | 1.764 | ✗ | 1.773 | ✗ | 0.850 | ✗ |

*A.3.2 Supplementary experimental results on the Law School Dataset.* Table 4 shows the predictive performance (RMSE), ability parameter $\theta_i$, and estimated $\hat{STS}_i$ for 15 predictive models on the Law School dataset. We note that GBM_5 achieves the highest fairness performance. However, GBM_1 is the best model in predictive performance. Table 5 provides detailed results for selected individuals from the Law School dataset. The selected individuals are those with the five smallest flatness indicators.

**Table 4: The predictive performance (RMSE), ability parameter $\theta_i$, and estimated $\hat{STS}_i$ for 15 predictive models on the Law School dataset are presented. Note that $\hat{STS}_i$ represents the estimated STS for each predictive model, computed using Equation 7.**

| Model | RMSE | Ability | $\hat{STS}_i$ | Model | RMSE | Ability | $\hat{STS}_i$ |
|---|---|---|---|---|---|---|---|
| GBM_1 | 0.8632 | 0.9042 | 0.7014 | GBM_8 | 0.8653 | 0.8810 | 0.6432 |
| GBM_2 | 0.8635 | 0.8993 | 0.6686 | GBM_9 | 0.8664 | 0.8622 | 0.6364 |
| DP_1 | 0.8639 | 0.9102 | 0.6839 | DP_2 | 0.8679 | 0.9111 | 0.6762 |
| GBM_3 | 0.8648 | 0.8987 | 0.6812 | GBM_10 | 0.8684 | 0.8346 | 0.5914 |
| GBM_4 | 0.8648 | 0.8876 | 0.6496 | GBM_11 | 0.8771 | 0.5644 | 0.3873 |
| GBM_5 | 0.8650 | 0.9239 | 0.7571 | DRF_1 | 0.8846 | 0.6119 | 0.4243 |
| GBM_6 | 0.8651 | 0.8669 | 0.6183 | XRT_1 | 0.8862 | 0.9340 | 0.7507 |
| GLM_7 | 0.8652 | 0.9072 | 0.6611 | | | | |

*A.3.3 Supplementary experimental results with different sensitivity attributes on the Adult Dataset.* In this section, we continue using the Adult dataset but set *race* as the sensitive attribute. We simulate 14 predictive models for the Adult dataset. Figure 6(a) shows the scatter plot of the discrimination parameter $a_j$ and the difficulty parameter $\delta_j$ for each individual in the evaluation set. The purple dot

**Table 5: The quantitative way of disentangling selected individuals from the Law School dataset.** ✓ indicates that the individual is treated fairly under the selected predictive model, while ✗ indicates the opposite.

| Model | $\log \Theta_i$ | Individual 539 ($\log \Delta_j$=4.438) $g(\hat{STS}_{ij})$ | | Individual 219 ($\log \Delta_j$=4.258) $g(\hat{STS}_{ij})$ | | Individual 920 ($\log \Delta_j$=3.872) $g(\hat{STS}_{ij})$ | | Individual 356 ($\log \Delta_j$=4.738) $g(\hat{STS}_{ij})$ | | Individual 572 ($\log \Delta_j$=-3.048) $g(\hat{STS}_{ij})$ | |
|---|---|---|---|---|---|---|---|---|---|---|---|
| GBM_1 | -1.737 | 2.701 | ✗ | 2.521 | ✗ | 2.135 | ✗ | 3.001 | ✗ | -4.786 | ✓ |
| GBM_2 | -1.586 | 2.852 | ✗ | 2.672 | ✗ | 2.286 | ✗ | 3.152 | ✗ | -4.635 | ✓ |
| DP_1 | -1.670 | 2.769 | ✗ | 2.589 | ✗ | 2.203 | ✗ | 3.069 | ✗ | -4.718 | ✓ |
| GBM_3 | -1.585 | 2.853 | ✗ | 2.673 | ✗ | 2.287 | ✗ | 3.153 | ✗ | -4.633 | ✓ |
| GBM_4 | -1.409 | 3.030 | ✗ | 2.849 | ✗ | 2.463 | ✗ | 3.329 | ✗ | -4.457 | ✓ |
| GBM_5 | -2.361 | 2.078 | ✗ | 1.897 | ✗ | 1.511 | ✗ | 2.377 | ✗ | -5.409 | ✓ |
| GBM_6 | -1.106 | 3.333 | ✗ | 3.153 | ✗ | 2.767 | ✗ | 3.633 | ✗ | -4.154 | ✓ |
| GLM_7 | -1.567 | 2.872 | ✗ | 2.692 | ✗ | 2.306 | ✗ | 3.172 | ✗ | -4.615 | ✓ |
| GBM_8 | -1.315 | 3.124 | ✗ | 2.943 | ✗ | 2.557 | ✗ | 3.423 | ✗ | -4.363 | ✓ |
| GBM_9 | -1.244 | 3.194 | ✗ | 3.014 | ✗ | 2.628 | ✗ | 3.494 | ✗ | -4.293 | ✓ |
| DP_2 | -1.524 | 2.914 | ✗ | 2.734 | ✗ | 2.348 | ✗ | 3.214 | ✗ | -4.573 | ✓ |
| GBM_10 | -0.897 | 3.541 | ✗ | 3.361 | ✗ | 2.975 | ✗ | 3.841 | ✗ | -3.945 | ✓ |
| GBM_11 | 0.469 | 4.908 | ✗ | 4.728 | ✗ | 4.342 | ✗ | 5.208 | ✗ | -2.579 | ✓ |
| DRF_1 | 0.298 | 4.736 | ✗ | 4.556 | ✗ | 4.170 | ✗ | 5.036 | ✗ | -2.751 | ✓ |
| XRT_1 | -2.461 | 1.977 | ✗ | 1.797 | ✗ | 1.411 | ✗ | 2.277 | ✗ | -5.509 | ✓ |

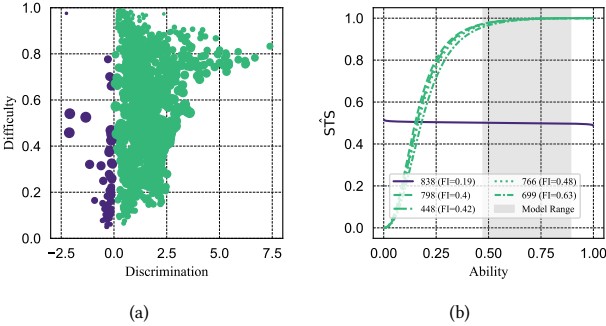

(a) (b)

**Figure 6: The plots for the Adult dataset with *race* the sensitive attribute: (a) The scatter plot shows the discrimination parameter $a_j$ and the difficulty parameter $\delta_j$ for each individual; (b) Examples of selected individuals with flat ICCs. The shaded area indicates the ability range of the 14 selected predictive models.**

denotes an individual identified as a special case, where the value of the discrimination parameter is negative. Figure 6(b) presents the ICCs for individuals with the five smallest flatness indicators. Table 6 provide detailed results for selected individuals from the Adult dataset with *race* as the sensitivity attribute. The selected individuals are those with the five smallest flatness indicators. These results demonstrate that the proposed Fair-IRT framework is not restricted to the sensitive attribute and has generalisation ability.

**Table 6: The quantitative way of disentangling selected individuals from the Adult dataset with *race* as the sensitivity attribute. ✓ indicates that the individual is treated fairly under the selected predictive model, while ✗ indicates the opposite.**

| Model | $\log \Theta_i$ | Individual 838 ($\log \Delta_j$=1.489) $g(\hat{STS}_{ij})$ | | Individual 798 ($\log \Delta_j$=-3.097) $g(\hat{STS}_{ij})$ | | Individual 448 ($\log \Delta_j$=-3.018) $g(\hat{STS}_{ij})$ | | Individual 766 ($\log \Delta_j$=-3.093) $g(\hat{STS}_{ij})$ | | Individual 699 ($\log \Delta_j$=-2.882) $g(\hat{STS}_{ij})$ | |
|---|---|---|---|---|---|---|---|---|---|---|---|
| GBM_2 | -1.823 | -0.334 | ✓ | -4.920 | ✓ | -4.841 | ✓ | -4.916 | ✓ | -4.705 | ✓ |
| GBM_5 | -1.721 | -0.232 | ✓ | -4.818 | ✓ | -4.739 | ✓ | -4.814 | ✓ | -4.603 | ✓ |
| GBM_3 | -1.832 | -0.343 | ✓ | -4.928 | ✓ | -4.850 | ✓ | -4.925 | ✓ | -4.714 | ✓ |
| GBM_4 | -1.604 | -0.115 | ✓ | -4.701 | ✓ | -4.622 | ✓ | -4.697 | ✓ | -4.486 | ✓ |
| GBM_g4 | -1.443 | 0.046 | ✗ | -4.539 | ✓ | -4.461 | ✓ | -4.536 | ✓ | -4.325 | ✓ |
| GBM_g2 | -1.621 | -0.132 | ✓ | -4.718 | ✓ | -4.639 | ✓ | -4.714 | ✓ | -4.503 | ✓ |
| GBM_g3 | -1.574 | -0.085 | ✓ | -4.671 | ✓ | -4.592 | ✓ | -4.667 | ✓ | -4.456 | ✓ |
| GBM_1 | -1.812 | -0.323 | ✓ | -4.908 | ✓ | -4.830 | ✓ | -4.905 | ✓ | -4.694 | ✓ |
| GBM_g1 | -1.363 | 0.126 | ✗ | -4.460 | ✓ | -4.381 | ✓ | -4.456 | ✓ | -4.245 | ✓ |
| GBM_g5 | -1.579 | -0.090 | ✓ | -4.676 | ✓ | -4.597 | ✓ | -4.672 | ✓ | -4.461 | ✓ |
| XRT_1 | -2.171 | -0.682 | ✓ | -5.267 | ✓ | -5.188 | ✓ | -5.263 | ✓ | -5.052 | ✓ |
| DRF_1 | -1.076 | 0.413 | ✗ | -4.173 | ✓ | -4.094 | ✓ | -4.169 | ✓ | -3.958 | ✓ |
| DL_1 | -1.135 | 0.354 | ✗ | -4.232 | ✓ | -4.153 | ✓ | -4.228 | ✓ | -4.017 | ✓ |
| GLM_1 | 0.391 | 1.880 | ✗ | -2.706 | ✓ | -2.627 | ✓ | -2.702 | ✓ | -2.491 | ✓ |

**Table 7: The quantitative way of disentangling selected individuals from the Adult dataset with ES as the fairness metrics. ✓ indicates that the individual is treated fairly under the selected predictive model, while ✗ indicates the opposite.**

| Model | $\log \Theta_i$ | Individual 971 ($\log \Delta_j$=-3.768) $g(\hat{ES}_{ij})$ | | Individual 798 ($\log \Delta_j$=-3.557) $g(\hat{ES}_{ij})$ | | Individual 756 ($\log \Delta_j$=1.577) $g(\hat{ES}_{ij})$ | | Individual 448 -3.380 $g(\hat{ES}_{ij})$ | | Individual 766 ($\log \Delta_j$=-2.882) $g(\hat{ES}_{ij})$ | |
|---|---|---|---|---|---|---|---|---|---|---|---|
| GBM_3 | -1.681 | -5.449 | ✓ | -5.238 | ✓ | -0.104 | ✓ | -5.061 | ✓ | -5.168 | ✓ |
| GBM_2 | -1.641 | -5.410 | ✓ | -5.199 | ✓ | -0.064 | ✗ | -5.021 | ✓ | -5.129 | ✓ |
| GBM_5 | -1.562 | -5.331 | ✓ | -5.120 | ✓ | 0.015 | ✗ | -4.942 | ✓ | -5.050 | ✓ |
| GBM_4 | -2.113 | -5.881 | ✓ | -5.670 | ✓ | -0.535 | ✓ | -5.493 | ✓ | -5.600 | ✓ |
| GBM_g4 | -1.569 | -5.337 | ✓ | -5.126 | ✓ | 0.009 | ✗ | -4.949 | ✓ | -5.056 | ✓ |
| GBM_g2 | -1.048 | -4.817 | ✓ | -4.606 | ✓ | 0.529 | ✗ | -4.428 | ✓ | -4.536 | ✓ |
| GBM_1 | -2.026 | -5.795 | ✓ | -5.584 | ✓ | -0.449 | ✓ | -5.406 | ✓ | -5.514 | ✓ |
| GBM_g3 | -1.396 | -5.164 | ✓ | -4.953 | ✓ | 0.181 | ✗ | -4.776 | ✓ | -4.883 | ✓ |
| GBM_g1 | -1.375 | -5.144 | ✓ | -4.933 | ✓ | 0.202 | ✗ | -4.755 | ✓ | -4.863 | ✓ |
| GBM_g5 | -2.946 | -6.714 | ✓ | -6.503 | ✓ | -1.369 | ✓ | -6.326 | ✓ | -6.433 | ✓ |
| XRT_1 | -2.869 | -6.637 | ✓ | -6.426 | ✓ | -1.291 | ✓ | -6.249 | ✓ | -6.356 | ✓ |
| DRF_1 | -0.937 | -4.705 | ✓ | -4.494 | ✓ | 0.641 | ✗ | -4.317 | ✓ | -4.424 | ✓ |
| DL_1 | -0.799 | -4.568 | ✓ | -4.357 | ✓ | 0.778 | ✗ | -4.179 | ✓ | -4.287 | ✓ |
| GLM_1 | 1.225 | -2.543 | ✓ | -2.332 | ✓ | 2.803 | ✗ | -2.155 | ✓ | -2.262 | ✓ |

Received 20 February 2007; revised 12 March 2009; accepted 5 June 2009

*A.3.4 Supplementary experimental results with different fairness metrics on the Adult Dataset.* In this section, we continue using the Adult dataset but set Equalised Score (ES) as the fairness metric. We simulate 14 predictive models for the Adult dataset. Figure 7(a) shows the scatter plot of the discrimination parameter $a_j$ and the difficulty parameter $\delta_j$ for each individual in the evaluation set. The purple dot denotes an individual identified as a special case, where the value of the discrimination parameter is negative. Figure 7(b) presents the ICCs for individuals with the five smallest flatness indicators. Table 7 provide detailed results for selected individuals from the Adult dataset. The selected individuals are those with the five smallest flatness indicators. These results demonstrate that the proposed Fair-IRT framework is suitable for different fairness metrics.

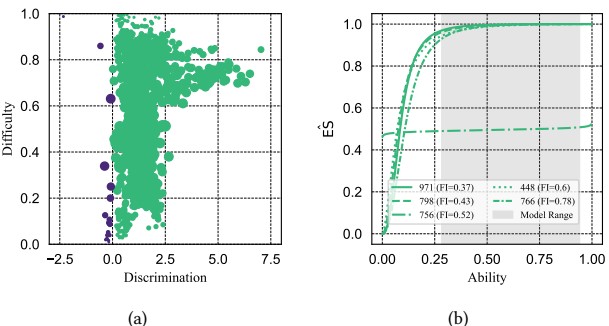

(a)     (b)

**Figure 7: The plots for the Adult dataset with *race* the sensitive attribute: (a) The scatter plot shows the discrimination parameter $a_j$ and the difficulty parameter $\delta_j$ for each individual; (b) Examples of selected individuals with flat ICCs. The shaded area indicates the ability range of the 14 selected predictive models.**

