# OpenReview forum: "Fairness Evaluation with Item Response Theory"
_ACM.org/TheWebConf/2025/Conference — WWW 2025 Poster_

### Official Review · Reviewer_qMH2 · 2024-11-28

**Novelty:** 5
**Technical Quality:** 4

**Review:**

Summary: This paper presents a method to evaluate fairness over a set of ML models, using the "Item Response Theory" (IRT) framework from psychometrics. They make an analogy with ML models as test respondents, and individuals in a dataset as test items. The primary distinction the authors make between this work and previous work is that they evaluate the *success* of a model in terms of its ability to score each individual *fairly*. The notion of fairness they apply in the paper is "Situation Test Score" (STS), which is the sensitivity of the model's scores to a protected attribute $A$.

The main contributions are:
- Methodological: they outline a "workflow" through which the IRT method can evaluate fairness over a set of models.
- Empirical: They demonstrate their workflow on a simulated set of models, and on the Adult and Law School datasets.

Strengths:
- It is interesting to see the IRT framework applied to evaluating fairness. The method of inferring item difficulty and model quality from looking at a set of models is a nice application of this IRT methodology to machine learning. The use of this methodology for fairness also seems promising, provided a fairness metric can actually be computed for an individual.
- The paper provides nice coverage of related work and background on IRT methods from psychometrics.
- The highlighting of model ability, item difficulty, and item discrimination parameters is an interesting way to analyze a set of models.

Weaknesses:
1. The choice of the set of models evaluated seems to be extremely important to the inferred model ability and difficulty parameters. The paper does not state or discuss this in enough detail. I suggest that the authors add a more in-depth discussion of potential issues that can arise in choosing the set of models to be evaluated, and how that selection process can affect the validity or generalizability of the estimated parameters, $\theta, \delta, \alpha$.

1a. For instance, the initial set of models seems to be key to the process of disentangling "the unfairness between individuals and predictive models," where the unfairness "due to the individual" is really just a property of the set of models. Under what conditions would the individual still exhibit the same "difficulty" beyond the initial given set of models?

2. Assumption 1 is quite coarse: it only says that the the set of models will have different fairness levels per individual. Can the authors more clearly delineate (in technical terms) how this assumption makes it possible to estimate the $\theta, \delta, \alpha$ parameters?

3. There is not enough detail about the estimation of the $\theta, \delta, \alpha$ parameters. Algorithm 1 in the Appendix is too vague. What is the "loss" defined as on Line 1064? What is the "3-parameter beta IRT" procedure exactly on Line 1063? What are the convergence or sample complexity properties of this estimation procedure? I could not find this discussed anywhere in the main paper or the appendix.

4. There is not enough detail about the computation/estimation of $STS_{ij}$. Are the expectations and probabilities in (5) and (6) estimated from the data? I did not see a precise description of how this is estimated in the main paper or the appendix. There is a formulation of $\hat{STS}$ based on $\theta, \delta, \alpha$ parameters (as in Equation (8)), but step iii in the workflow on Line 380 says that $STS_{ij}$ is computed first before $\theta, \delta, \alpha$ are estimated. So, how is $STS_{ij}$ computed in step iii?

5. The primary fairness measure studied, STS, is not the most useful for machine learning models. This notion is equivalent to "fairness through unawareness," which is both trivial to satisfy in machine learning models (simply train the model omitting a given feature), and not always practically or legally the most relevant notion (see e.g. https://arxiv.org/abs/1104.3913). It would be useful to have the authors demonstrate their method using a different notion of fairness, such as the definitions supplied in Appendix A.2.

6. The extensions of the method to demographic parity, equal opportunity, and equal odds in Appendix does not seem meaningful to me. Appendix states that "Demographic parity is a restricted version of the situation test score." This perhaps true for the version of demographic parity written on Line 1124, but I don't think this is a good generalization of demographic parity. There is a significant difference between requiring that $P(\hat{Y} | A = 0) = P(\hat{Y} | A = 1)$ and $P(\hat{Y} | A = 0, X = x) = P(\hat{Y} | A = 1, X = x)$. Satisfying the second version can still yield arbitrary violations in the first version, depending on the distribution of $X$, hence why "fairness through blindness" is not enough (see also https://arxiv.org/abs/1104.3913). The equal odds extension also seems to trivially reduce to the STS/demographic parity statement on Line 1124, since for a machine learning model $\hat{Y}$ which is a function of $X$, $\hat{Y}$ is conditionally independent of $Y$ given $X$, so satisfying $P(\hat{Y} | A = 0, X = x) = P(\hat{Y} | A = 1, X = x)$ implies that $P(\hat{Y} | A = 0, X = x, Y = y) = P(\hat{Y} | A = 1, X = x, Y = y)$, and vice versa. Overall, I suggest that the authors revisit these extensions.

**Questions:**

See Weaknesses section above.

**Reviewer Confidence:**

3: The reviewer is confident but not certain that the evaluation is correct

**Scope:**

3: The work is somewhat relevant to the Web and to the track, and is of narrow interest to a sub-community

---

### Official Review · Reviewer_HVTu · 2024-11-30

**Novelty:** 4
**Technical Quality:** 4

**Review:**

The paper introduces the innovative Fair-IRT framework, which applies Item Response Theory (IRT) to fairness evaluation for the first time. The dual methods for disentangling unfairness sources—flatness analysis of Item Characteristic Curves (ICCs) and the quantitative Rasch Beta IRT model—are unique contributions, providing new perspectives for fairness analysis.

Strength：
1. The paper combines rigorous theoretical derivations with practical experiments. It effectively bridges IRT concepts (e.g., ability, difficulty, discrimination) with fairness metrics (e.g., STS), offering interpretable and actionable insights.
2. The paper combines rigorous theoretical derivations with practical experiments. It effectively bridges IRT concepts (e.g., ability, difficulty, discrimination) with fairness metrics (e.g., STS), offering interpretable and actionable insights.
3. The paper provides a fresh lens for fairness evaluation, contributing to more inclusive and trustworthy AI systems.

Weakness：
1. The framework relies on a single fairness metric (STS). Incorporating other commonly used fairness metrics (e.g., demographic parity, equalized odds) or multi-dimensional fairness analysis would enhance its robustness.
2.  Only two datasets (Adult and Law School) were used in the study, both focusing on income prediction or academic performance with gender as the sensitive attribute. The applicability of Fair-IRT in other domains, such as NLP or computer vision, remains unclear, and the exploration of additional influencing factors is insufficient.

**Questions:**

1. The threshold for fairness ( 𝜖 = 0.5 ) is consistent across experiments. How was this value determined, and does it generalize well to other datasets or fairness metrics? Would dynamic thresholding improve results?
2. The paper identifies individuals with negative discrimination as “special cases.” Have these individuals been examined for potential data issues (e.g., outliers, imbalances) that might skew results?

**Ethics Review Flag:**

Yes

**Reviewer Confidence:**

3: The reviewer is confident but not certain that the evaluation is correct

**Scope:**

3: The work is somewhat relevant to the Web and to the track, and is of narrow interest to a sub-community

---

### Official Review · Reviewer_cstd · 2024-12-02

**Novelty:** 7
**Technical Quality:** 7

**Review:**

## My interpretation of the work's key contributions and approach.

This paper presents a novel application of Item Response Theory (IRT) to the field of AI fairness evaluation: while IRT has been traditionally used in psychometrics to assess student abilities and test question characteristics, the authors demonstrate its potential for evaluating fairness in machine learning models. In the proposed framework, Fair-IRT, which builds upon the beta IRT, predictive models serve as respondents when making decisions regarding individuals - such as hiring decisions or salary recommendations - while people in the evaluation set serve as items, as the ultimate goal is to evaluate decision fairness. A key contribution of Fair-IRT is its ability to disentangle sources of unfairness. The framework accomplishes this through two complementary approaches: First, it analyzes the flatness of Item Characteristic Curves (ICCs), where flat curves typically indicate unfairness stemming from individual characteristics rather than model bias - since that even models with higher fairness scores do not discriminate less for them -,Second, it employs a quantitative decomposition based on the Rasch beta IRT model, which explicitly separates the contributions of individual characteristics and model bias to observed unfairness. The authors validate their framework through both theoretical analysis using simulated data to establish clear interpretations of the model parameters, and practical applications to two real-world datasets, demonstrating its effectiveness in identifying both problematic models and individuals who may face systematic unfairness.

## Quality, clarity, originality and significance of this work.

This paper shows good quality in its methodological approach. The authors develop systematically their framework, that is well-grounded in established theoretical foundations while introducing novel applications. The mathematical formulations are well-defined, and the assumptions underlying their approach are explicitly stated and justified. The paper also shows attention to reproducibility by providing their source code.I particularly appreciate their validation strategy that combines simulated data with real-world datasets, though I think more extensive testing on different types of datasets would make their findings more robust.
The paper's structure and presentation are generally clear. The authors explain IRT concepts progressively and support technical content with helpful visualizations. However, I find that some crucial mathematical derivations, especially in the beta IRT model specifications, could benefit from more detailed explanations. WWhile some sections, particularly those dealing with beta IRT model specifications, require careful reading, the authors support these technical details with helpful interpretations and visual aids.
In terms of originality, the paper presents a new application of IRT to the field of AI fairness. Based on my literature search and to the best of my knowledge, I can confirm the authors' claim that this is the first work to use IRT models for evaluating fairness in both predictive models and individuals.
The significance of this work lies in its potential practical applications, though I have some reservations. While the framework shows promise in identifying sources of unfairness, I believe it needs more validation. Specifically, the authors should have included comparisons with existing fairness evaluation methods to demonstrate concrete advantages of their approach.
=

## Pros

1. The novelty of applying IRT to fairness evaluation in Machine Learning represents an innovative contribution to this field.
2. The framework provides a systematic way to evaluate how different models balance accuracy and fairness. This makes it a valuable tool for real-world applications, where organizations need to understand the fairness implications of their model choices while maintaining required performance levels.
3. One of the framework's most significant contributions is its ability to disentangle sources of unfairness, helping distinguish whether unfair treatment stems from model bias or individual characteristics. Moreover, another key strength is that it can work with various fairness measures.

## Cons

1. While effective for evaluating fairness, this approach maintains machine learning models as "black boxes." By treating fairness ability as a latent variable, we can observe and measure unfairness but cannot directly address its root causes within the model architecture. Though useful for model selection by identifying which models discriminate least, it doesn't provide insights into how to modify model parameters to improve fairness. It might be possible to gain some insights through reverse engineering - observing how modifications to model structures affect fairness behaviors - but this remains complex and far from the current need for explainable AI.

2. The framework relies on a strict assumption about model diversity, requiring that predictive models exhibit "a diversity of fairness performance" with "sufficiently sparse" values. This assumption might limit the framework's applicability in scenarios where we want to evaluate a set of models that are all designed to be fair and determine which achieves the highest fairness.

3. The authors currently implement and validate their framework using a single sensitive attribute, and they thoughtfully acknowledge this as a limitation since real-world scenarios often involve multiple interacting attributes. While this affects the immediate applicability in some practical situations, the authors have already identified this as a promising direction for future research, demonstrating their awareness of the framework's potential for expansion.

**Questions:**

### Interpretation of Individual-Based Unfairness
The paper's analysis of unfairness stemming from individuals warrants deeper examination. While the framework effectively identifies different patterns of individual-based unfairness, I believe the paper would benefit from more concrete examples and detailed analysis.
The first case involves individuals with high difficulty parameters (δⱼ → 1), who experience unfair treatment systematically from most models, except those with very high fairness ability. While the framework effectively identifies these cases, I struggle to envision real-world examples of what non-sensitive characteristics might lead to such systematic unfairness. Interestingly, looking at Figure 4(a), individuals with difficulty parameters approaching 1 seem to also exhibit low discrimination values (0 < aⱼ < 1). This clustering pattern raises an interesting question: do these individuals share common features that might explain both their high difficulty and low discrimination characteristics?
This observation leads to the second case of individuals exhibiting flat ICC regions (0 < aⱼ < 1). Given the demonstrated range of model fairness abilities, understanding more clearly why certain individuals receive consistent treatment regardless of model capability could reveal important patterns in systematic unfairness.
The third and most counterintuitive case involves individuals with negative discrimination parameters (aⱼ < 0). Simply flagging these as "special cases" requiring further investigation feels insufficient, as they challenge the framework's fundamental logic by suggesting that more fair models produce unfairer predictions. Once more, a qualitative description of these individuals' characteristics (shown as purple dots in Figure 4(a)) would help the reader to understand what makes these cases truly "special.
### Assessment of Interdependent Decisions
The Fair-IRT framework treats each individual's evaluation independently when assessing model fairness. However, this assumption may not hold in many real-world scenarios where decisions are inherently interconnected. For example, in hiring processes, organizations often work with fixed quotas or limited positions, making decisions about one candidate directly impact the opportunities available to others. Similarly, in university admissions or loan approvals, decisions are often made comparatively within a pool of candidates rather than in isolation.
This raises important questions about how such dependencies might affect Fair-IRT's ability to disentangle sources of unfairness. When an unfair decision for one individual directly influences decisions for others, can the framework still effectively distinguish between model bias and individual characteristics? Moreover, how might these interconnected decisions affect the interpretation of the discrimination and difficulty parameters?

**Reviewer Confidence:**

3: The reviewer is confident but not certain that the evaluation is correct

**Scope:**

3: The work is somewhat relevant to the Web and to the track, and is of narrow interest to a sub-community

---

### Official Review · Reviewer_XGSX · 2024-12-03

**Novelty:** 5
**Technical Quality:** 5

**Review:**

This work presents Fair-IRT, a novel framework for evaluating the fairness performance of predictive models and individuals using Item Response Theory (IRT). This is the first application of IRT to fairness evaluation.

Originality and Significance:
- The paper introduces an interesting and, to my knowledge, a new approach to fairness evaluation by applying IRT, a concept traditionally used in psychometrics, to machine learning.
- It provides a comprehensive evaluation of both models and individuals, which allows for a deeper understanding of how fairness is manifested and how to mitigate biases.
- The authors provide a detailed explanation of the Fair-IRT framework, including its workflow, the interpretation of its parameters, and its application to real-world datasets.

Pros:
- The paper is well-organized and presents its ideas clearly and concisely. The authors provide sufficient background information on IRT and fairness evaluation, making the paper accessible to a wide audience.
- Fair-IRT evaluates fairness from two perspectives. The first is the individual level, which identifies "special individuals" treated unfairly regardless of the model's fairness ability. The second is the Model Level, which assesses the fairness ability of different predictive models, allowing for ranking and comparison based on their ability to make fair predictions.
- The authors demonstrate the framework's applicability to different tasks (classification and regression), sensitive attributes, and fairness metrics.

Cons:
- The current version of Fair-IRT is limited to evaluating fairness based on a single metric and sensitive attribute.
- The quantitative disentanglement method, which utilizes the Rasch beta IRT model, may be less suitable for complex scenarios due to the model's lower fitting power compared to the original beta IRT model.
- The framework relies on the assumption that the set of predictive models being evaluated exhibits a diverse range of fairness performances. While practical, this assumption might not hold in all cases and could limit the framework's applicability.

**Questions:**

The current Fair-IRT framework focuses on a single fairness metric at a time. How would the framework be extended to incorporate multiple fairness metrics simultaneously? Would a multi-dimensional IRT model be suitable for this purpose?

Given the variety of fairness metrics available, how does the choice of a specific fairness metric influence the interpretation of Fair-IRT results? Are there any guidelines for selecting the most appropriate metric for a given task and dataset?

While the paper acknowledges the limited fitting power of the Rasch beta IRT model used in the quantitative disentanglement method, could you elaborate on the types of scenarios where this approach might be less effective?

The framework assumes a diverse range of fairness performances across the set of predictive models being evaluated. How can practitioners ensure this assumption is met in real-world applications? What are the potential consequences if this assumption is violated?

Can you provide concrete examples of how the framework could be used to guide the development of fairer models or mitigate biases in existing systems?

The paper mentions the potential of Fair-IRT to evaluate fairness-aware predictive models. Could you expand on how this evaluation would differ from evaluating non-fairness-aware models? What specific insights could be gained from applying Fair-IRT to fairness-aware models?

**Reviewer Confidence:**

2: The reviewer is willing to defend the evaluation, but it is likely that the reviewer did not understand parts of the paper

**Scope:**

3: The work is somewhat relevant to the Web and to the track, and is of narrow interest to a sub-community